# Research Progress in Land Consolidation and Rural Revitalization: Current Status, Characteristics, Regional Differences, and Evolution Laws

Shuchang Li [1,2] and Wei Song [1,3,*]

1 Key Laboratory of Land Surface Pattern and Simulation, Institute of Geographic Sciences and Natural Resources Research, Chinese Academy of Sciences, Beijing 100101, China
2 University of Chinese Academy of Sciences, Beijing 100049, China
3 Hebei Coordinated Innovation Center for Urban-Rural Integration Development, Shijiazhuang 050061, China
* Correspondence: songw@igsnrr.ac.cn

**Abstract:** As an invaluable tool to manage land use, land consolidation has been widely used globally, promoting rural revitalization in the context of the current global rural decline. A comprehensive analysis of land consolidation and rural revitalization will help to promote rural development and poverty alleviation and achieve the goals of rural revitalization and regional sustainable development. Based on publications on land consolidation and rural revitalization from 1950 to 2021 in the Web of Science database, this study analyzes the research status, characteristics, regional divides, and evolution laws in land consolidation and rural revitalization using the Bibliometrix and Biblioshiny software packages. The results are as follows: (1) The number of publications per year on land consolidation and rural revitalization increased. According to the publication number, this period can be divided into three stages: the initial stage from 1950 to 2000, the growth stage from 2001 to 2012, and the high-production stage from 2013 to 2021. (2) From 1950 to 2021, 1715 authors published papers on land consolidation and rural revitalization. (3) Respective studies were published by authors in 89 countries or regions, mainly China, the United States, and Poland. Of these, China and the United States played an important role in international cooperation. (4) The keywords in land consolidation and rural revitalization were related to (i) technical efficiency and agricultural production, (ii) the impacts of policy management and systems, and land fragmentation; (iii) the impacts of land consolidation on land use transition against the background of urbanization; (iv) the evaluation of land consolidation. (5) Research on land consolidation has evolved from management over methods and models to land consolidation and regional consolidation.

**Keywords:** bibliometrics; Bibliometrix; land consolidation; rural revitalization; review





## 1. Introduction

Since the middle of the 20th century, land use management [1–4] and rural development [5–8] have gradually become major topics in science and politics, with an emphasis on land consolidation and rural revitalization [9,10]. With rapid urbanization, an increasing number of people migrate to cities, and the decline of the rural population has become a global trend [11,12]. Currently, more than 50% of the global population lives in urban areas, and by 2050, this number is expected to reach 66% [13,14]. Such a large urban population will lead to a substantial increase in food demand. As a large amount of agricultural land has already been occupied due to urban expansion, cultivated land is in shortage, and this trend is expected to further continue in the future. Consequently, a more sustainable land management model is needed to provide products and services to meet the growing food demand [15], and land consolidation plays an increasingly prominent role. As a useful tool for land management [16–18], land consolidation is crucial to balance the occupation and replenishment of arable land [19,20], ensure national food security [21], and

promote rural revitalization [22,23]. To achieve these goals in a sustainable manner, it is of great significance to carry out quantitative analysis and research in land consolidation and rural revitalization.

In recent years, land consolidation has become a major tool to ensure food security and promote rural revitalization. However, the connotation and function of land consolidation have undergone a long evolution. The term "land consolidation" was first proposed in Germany in 1343 and was later widely used in the Netherlands, Russia, and other countries [24–26]. Different countries use slightly different terms [27], such as "land improvement" in Japan [28,29] and "agriculture land redrawing" in Taiwan and China [30–32], but the essence and main content are the same [33]. Traditionally, land consolidation has been the most effective land management method for solving the problem of land fragmentation and improve land use efficiency [24,26,34,35]. With the development of the economy and society, land consolidation has been endowed with new goals and connotations [22,36]. Nowadays, land consolidation is a modern practice to improve land quality, natural landscapes, and the ecological environment by redeveloping and using land resources as well as promoting industrial transformation and development, thus improving people's income and quality of life and facilitating the comprehensive and sustainable development of the regional economy and society [24].

Against the background of global rural decline, land consolidation, a major tool to stabilize the dynamic balance of cultivated land and ensure food security, has been enriched and empowered with a new mission to promote rural revitalization [37]. Land consolidation aims at the rational development, use, governance, and protection of land resources and facilitates the management of rural land through means of planning, control, and engineering [38], thereby optimizing the allocation of rural land resources and promoting rural construction. Rural construction has been an important part of rural development for a long time, and land consolidation activities centered on the development, use, and transformation of land resources have always accompanied the construction and development of rural areas and have played a major role in increasing resource use efficiency, improving resource use conditions, and promoting rural development [39].

In the context of the COVID-19 pandemic and climate change, resource shortages, environmental degradation, and a lack of vitality have largely restricted rural revitalization and development. Land consolidation requires comprehensive research related to ecology, resources, society, the economy, and engineering technology [40] and can alleviate various issues by fostering drivers for rural revitalization. However, whilst land consolidation optimizes rural spatial layout and improves land use efficiency, it also causes some problems. On the one hand, land consolidation will aggravate land encroachment to a certain extent [41]. Although it solves the problem of cultivated land fragmentation, it may lead to the occupation of more cultivated land resources and the loss of the rural labor force, while will impede rural revitalization. On the other hand, land consolidation involves the ownership of land and the rights and interests of landowners [42]. In some situations, it is difficult to balance the interests of multiple landowners, which will lead to conflicts among various stakeholders. Therefore, if we want to ensure that land consolidation becomes a powerful tool to promote rural revitalization, we must solve the underlying issues.

With the increasing number of publications, the demand for new comprehensive research methods appeared in various fields [43]. As a tool that can analyze a large number of publications at both macro- and micro-levels, bibliometrics has become one of the major scientific research methods [44]. Pritchard [45] defined bibliometrics as the "application of mathematics and statistical methods to books and other media of communication", whereas Hawkins [46] defined it as "quantitative analysis in the bibliographic references of the body of literature". Historical bibliometrists recognized that adding the dimensions of time and space to bibliometric analysis can provide new insights into knowledge development [47,48]. Bibliometrics is a quantitative analysis method that uses mathematical and statistical tools to measure the relations and impacts of publications in specific research fields [3]. It enables researchers to draw complex knowledge maps representing the knowl-

edge structure of a certain research field and study their characteristics through statistical and mathematical methods [49,50]. As a powerful tool to analyze knowledge fields and reveal their cognitive structures [28,51], it provides a macro-overview of a large number of academic publications and convincingly identifies influential researchers, authors, journals, organizations, and countries [52]. Bibliometrix, as one of the major application software packages of bibliometrics, has assisted numerous scholars in literature analysis in various fields, enabling them to quickly grasp the research status, history, and future of the field [53–55].

Bibliometrics is getting increasingly popular [56,57] and has become one of the major analysis and research methods in geography [58,59]. In the field of land consolidation and rural revitalization, however, there are still few studies on visual analysis using bibliometrics. Some authors carried out a bibliometric analysis on land consolidation and rural revitalization through large-scale datasets and bibliometric methods, with valuable results [60]. For example, Shi Chendi et al. [61] summarized publications from the CNKI database in terms of "land consolidation" in the past 10 years from 2009 to 2018 from the perspective of core authors, core institutions, and journal rankings. Zang Yuzhu et al. [39], using a bibliometric approach, carried out quantitative and qualitative statistical analysis, cluster analysis, and main path analysis of publications in land consolidation published in the Web of Science database. However, most researchers focus on keyword co-occurrence, journal sources, and papers published by certain authors, whereas citations, clustering analysis of high-frequency keywords, discipline evolution, and the prediction of future development directions are largely ignored. Therefore, this paper systematically analyzes land consolidation articles published in the core collection database of Web of Science from 1950 to 2021 using the two R tools, Bibliometrix and Biblioshiny.

Current studies in this area mainly focus on the analysis of authors, journals, and keyword frequency but often ignore the analysis of citations, keyword clusters, and topic evolution, which can reflect the evolution and future trends of the field. Thus, based on 711 publications in the field of land consolidation and rural revitalization from 1950 to 2021 in the Web of Science database, we systematically sorted the number of articles, main countries/regions, main authors, and themes in this research field. The overall aim was to analyze the development status, characteristics, regional characteristics, and potential of land consolidation and rural revitalization, and the objectives were as follows:

(1) To analyze the keywords and their clusters and determine the evolution of the research topics and the focus of future research.
(2) To achieve cooperation in land consolidation and rural revitalization.
(3) To summarize the trends in the numbers of publications and citations in land consolidation and rural revitalization.

## 2. Data Source

Web of Science is the world's largest comprehensive academic information resource covering most disciplines; it includes more than 8700 core academic journals in fields such as natural sciences, engineering, biomedicine, social sciences, arts, and humanities. In this study, the search terms included TI = "land consolidation and rural revitalization" or TI = "land consolidation and village revitalization" or TI = "land remediation and village revitalization" and TI = "land reclamation and rural revitalization". The document type was "Article", the retrieval time was unlimited, the subject of the publications was unlimited, and the languages of the publications were English and Chinese. After preprocessing, such as the elimination of duplicates and irrelevant data, a total of 711 papers on land consolidation and rural revitalization were obtained. The data were downloaded in bib format.

## 3. Research Methods

The quantitative analysis software Bibliometrix was used to carry out qualitative and quantitative literature research on land consolidation and rural revitalization. By identifying past and current research topics, bibliometric analysis quantitatively and objectively provides a comprehensive analysis of a huge amount of publications [62]. The Bibliometrix R package is an open-source environment and system written in the R language, which is a useful tool for the quantitative analysis of scientific literature [63]. Compared with other languages, the R language has more efficient statistical algorithms, access to high-quality numerical equations, and integrable data visualization tools [64]. Biblioshiny was developed by Massimo Aria in the R language in the secondary development of the Bibliometrix-based shiny package [65,66]. It contains the core code of Bibliometrix and creates a framework for data analysis based on online databases [67]. Users can perform relevant scientific metrology and visual analysis on the interactive web interface [68,69], which, to a certain extent, reduces the use threshold and the information input intensity [70]. Bibliometrix boasts better visualizations, especially in keyword co-occurrence analysis [71]. Figure 1 shows the Bibliometrix workflow.

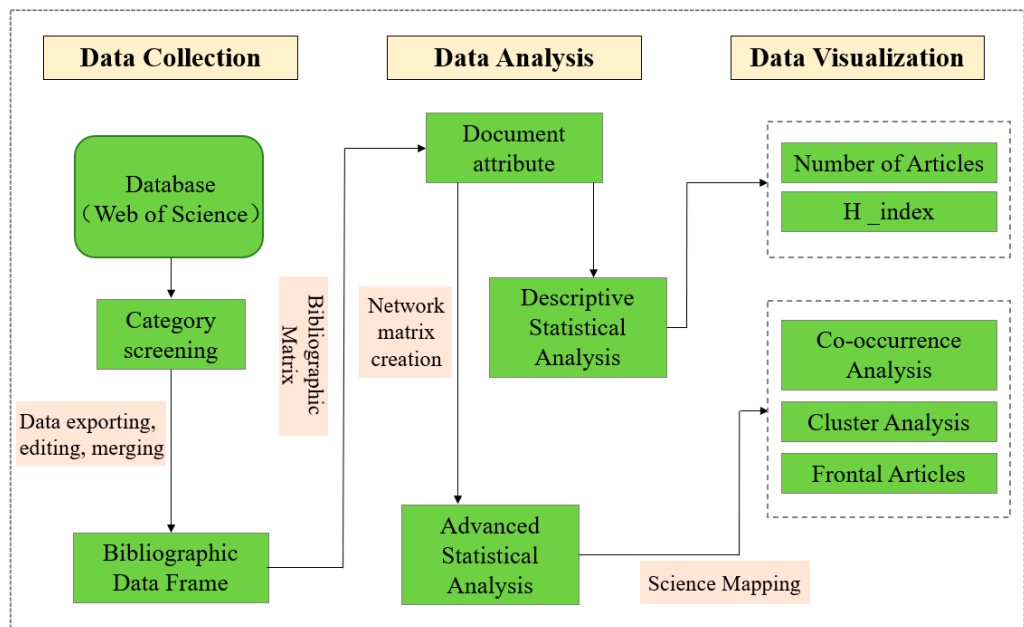

**Figure 1.** Science mapping workflow of Bibliometrix.

## 4. Result Analysis

### 4.1. Analysis of Publications

Analysis of the distribution of publications from the time series demonstrates the research trend (Figure 2). From 1950 to 2021, the number of publications on land consolidation and rural revitalization fluctuated slightly, with an overall increase. According to the changes in respective macro-policies, the period can be divided into three stages: from 2000 to 2012, and from 2013 to 2021. The period from 1950 to 2000 was the initial period, with a small number of publications per year. From 2001 to 2012, although the number of publications increased slowly, a stable state was reached, indicating that this field has attracted widespread attention. The period from 2013 to 2021 was the high production stage, with a rapidly increasing number of publications, peaking in 2018 with 93 articles.

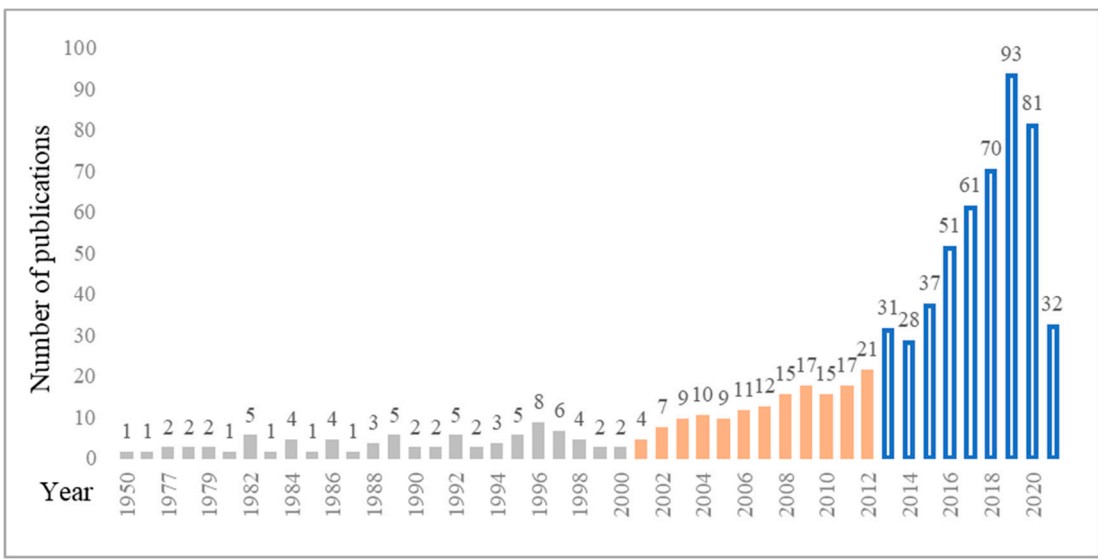

**Figure 2.** Average number of publications per year on land consolidation and rural revitalization from 1950 to 2021.

### 4.2. Author Analysis

A total of 1715 authors published articles on land consolidation and rural revitalization from 1950 to 2021. Among them, Liu Yansui, Li Yurui, and Long Hualou were the most productive ones (Figure 3) with 39, 25, and 16 papers, respectively. The h-index is not only a tool to measure the importance, significance, and impact of the authors' cumulative research contributions, but it can also be used to assess the current number of publications and predict an author's future performance as it combines productivity with impact. In this field, Liu Yansui from China, who published the most papers, had an h-index of 18, ranking first (Table 1), indicating that the publications of this author were of high quality and greatly impacted this research field. Both Liu Yansui and Long Hualou started their research in 2009. Liu's most cited paper was published in 2018 and Long's in 2014 (Figure 4).

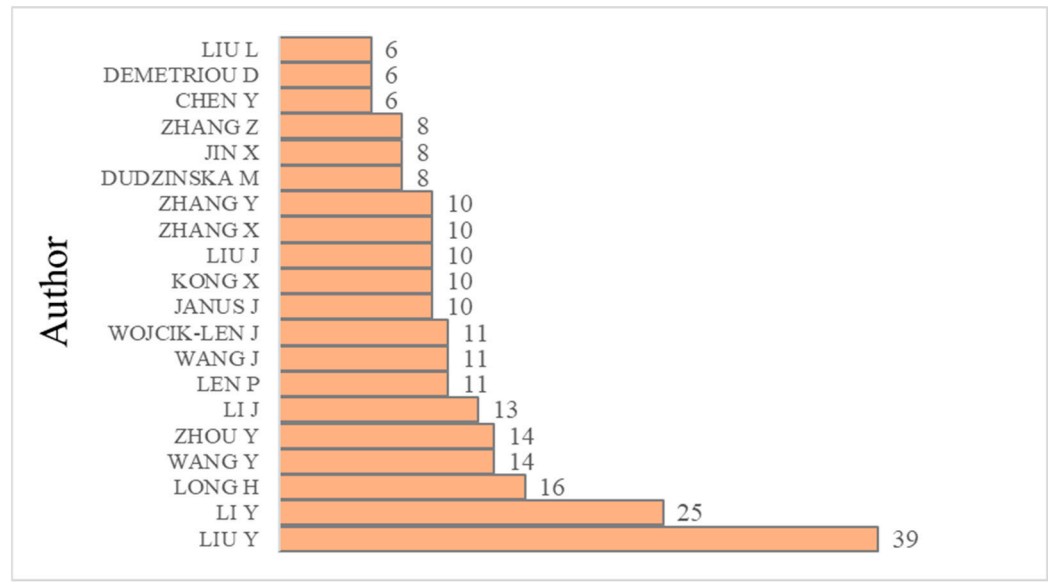

**Figure 3.** Top 20 authors publishing articles on land consolidation and rural revitalization from 1950 to 2021.

**Table 1.** Top 20 influential authors in the field of land consolidation and rural revitalization from 1950 to 2021.

| Author | H_Index | G_Index | M_Index | TC | NP | PY_Start |
|---|---|---|---|---|---|---|
| LIU Y | 18 | 32 | 1.286 | 2070 | 32 | 2009 |
| LI Y | 12 | 20 | 1.333 | 1131 | 20 | 2014 |
| LONG H | 12 | 14 | 0.857 | 1850 | 14 | 2009 |
| ZHOU Y | 10 | 13 | 1.25 | 481 | 13 | 2015 |
| KONG X | 8 | 9 | 1 | 216 | 9 | 2015 |
| WANG Y | 8 | 10 | 0.571 | 204 | 10 | 2009 |
| LI J | 7 | 10 | 0.875 | 450 | 10 | 2015 |
| ZHANG Y | 7 | 8 | 0.875 | 254 | 8 | 2015 |
| ZHANG Z | 7 | 8 | 0.636 | 201 | 8 | 2012 |
| JANUS J | 6 | 10 | 0.857 | 128 | 10 | 2016 |
| JIN X | 6 | 8 | 0.75 | 149 | 8 | 2015 |
| LONG HUALOU LH | 6 | 6 | 0.667 | 437 | 6 | 2014 |
| WANG J | 6 | 8 | 0.75 | 108 | 8 | 2015 |
| ZHANG X JIANG | 6 | 9 | 0.75 | 172 | 9 | 2015 |
| GUANGHUI JG | 5 | 5 | 0.625 | 168 | 5 | 2015 |
| LEN P | 5 | 10 | 0.714 | 106 | 10 | 2016 |
| LIU L | 5 | 6 | 0.833 | 112 | 6 | 2017 |
| LIU YANSUI LY | 5 | 5 | 0.5 | 271 | 5 | 2013 |
| TIAN Y | 5 | 5 | 0.714 | 93 | 5 | 2016 |
| CHEN W | 4 | 4 | 1 | 38 | 4 | 2019 |

Note: h-Index represents the importance and influence of the author's accumulated research; G-Index represents the derivative index of H index; M-Index = h/n, n represents the age of the author when he published his article in this field; TC stands for Total Citations; NP represents the number of articles; PY-Start stands for the initial publication time.

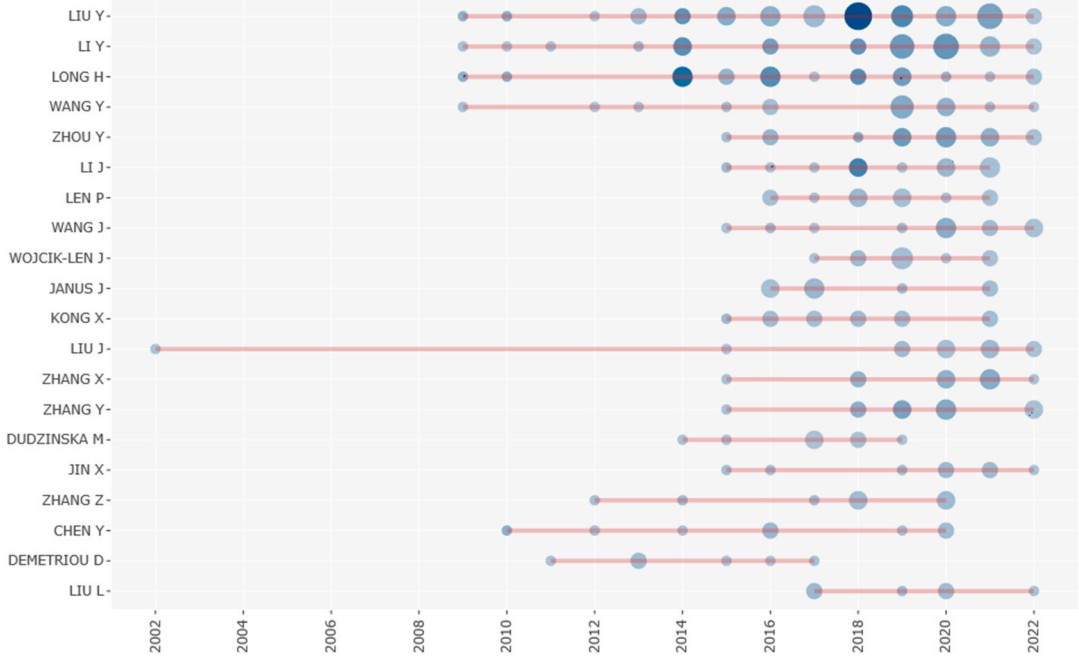

**Figure 4.** Numbers of publications and citations per year of the top 20 influential authors in the field of land consolidation and rural revitalization.

This section analyzes the number of publications and citations per year of the top 20 authors in land consolidation and rural revitalization. As seen in Figure 5, the most cited paper, published by Liu Yan in 2018, received 229 citations (the darkest spot) and was titled "Strategic adjustment of land use policy under the economic transformation", published in LAND USE POLICY. The year 2018 is also the year with the highest number of publications in this field. Both Li Yurui's and Long Hualou's most cited articles were published in 2014. Whilst Li Yurui published most articles in 2020, Long Hualou had the highest number of publications in 2016.

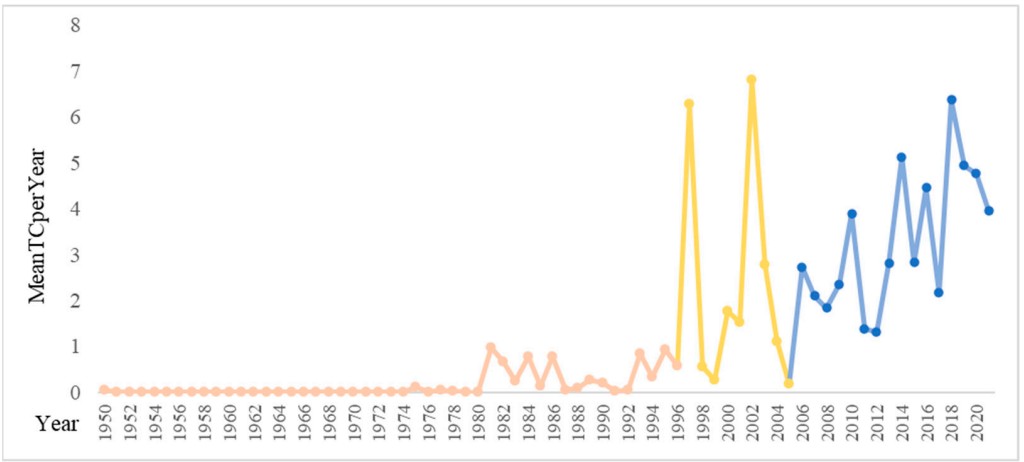

**Figure 5.** Average number of citations per year in land consolidation and rural revitalization from 1950 to 2021.

*4.3. Distribution Analysis for the Main Countries and Regions*

As seen in Table 2, most articles on land consolidation were published in Asia and Europe. China ranked first in the number of authors who published articles in this field and was the only developing country among the top three countries, accounting for 50% of the total output. This can be explained by the long history of land consolidation in China and the subsequent evolution of relevant policies. China was followed by Poland and the United States, with 164 and 141 articles, respectively. Although these countries had high outputs, their global influence was considerably lower than that of China. In general, China, Poland, and the United States had the greatest impacts on this research field and promoted efficient management strategies.

**Table 2.** Top 15 countries publishing articles in the field of land consolidation and rural revitalization from 1950 to 2021.

| Country | Number of Publications |
| --- | --- |
| CHINA | 1085 |
| USA | 164 |
| POLAND | 141 |
| UK | 93 |
| GERMANY | 55 |
| AUSTRALIA | 51 |
| SPAIN | 50 |
| CZECH REPUBLIC | 46 |
| INDIA | 46 |
| TURKEY | 42 |
| CANADA | 41 |
| NETHERLANDS | 34 |
| ITALY | 30 |
| BRAZIL | 23 |
| JAPAN | 23 |

According to Table 3, the countries cooperated closely. China and the United States were most involved in cooperations with other countries and were at the center of international cooperation in this field. China mainly cooperated with the United States, Australia, the United Kingdom, and Germany to publish 24, 10, 10, and 8 articles, respectively. The United States mainly cooperated with China, Brazil, Australia, and Canada to publish 24, 4, 3, and 3 articles, respectively. Among the 41 articles published in Canada, 5 were prepared in cooperation with China and 3 in cooperation with the United States. Among the 51 articles in Australia, 4 were published in cooperation with China and 3 in the United States.

**Table 3.** Main nations China cooperated within the field of land consolidation and rural revitalization from 1950 to 2021.

| From | To | Cooperation Frequency | From | To | Cooperation Frequency |
|---|---|---|---|---|---|
| CHINA | USA | 24 | CZECH REPUBLIC | SLOVAKIA | 3 |
| CHINA | AUSTRALIA | 10 | UNITED KINGDOM | CYPRUS | 3 |
| CHINA | UNITED KINGDOM | 10 | USA | AUSTRALIA | 3 |
| CHINA | GERMANY | 8 | USA | CANADA | 3 |
| CHINA | CANADA | 5 | AUSTRALIA | INDIA | 2 |
| CHINA | NETHERLANDS | 4 | AUSTRALIA | KENYA | 2 |
| UNITED KINGDOM | AUSTRALIA | 4 | AUSTRALIA | NETHERLANDS | 2 |
| USA | BRAZIL | 4 | AUSTRALIA | UGANDA | 2 |
| CHINA | BELGIUM | 3 | AUSTRALIA | VIETNAM | 2 |
| CHINA | JAPAN | 3 | CHINA | CZECH REPUBLIC | 2 |

*4.4. Citation Analysis*

4.4.1. Annual Development Trend of Citations

After analyzing the number of citations of papers in land consolidation and rural revitalization from 1950 to 2021 (Figure 5), we found that the citation rate from 1950 to 1996 was low, and from 1951 to 1979, the number of citations was zero, indicating that research in this field was in its infancy. The citation frequency averaged 2.64 from 1997 to 2004 and reached a peak of 6.81 in 2002. During this period, the citation frequency fluctuated more substantially than the publication number. From 2005 to 2021, the average citation rate increased, albeit with large fluctuations, while the number of papers published increased steadily.

Table 4 shows the 20 most cited journals in land consolidation and rural revitalization. Among them, LAND USE POLICY, J RURAL STUD, and HABITAT INT were leading, with 2932, 779, and 626 citations, respectively. The 2932 citations of the journal LAND USE POLICY make it the most cited and influential journal in this field. The journals NATURE and SCIENCE ranked 14 and 17, with 188 and 153 citations, respectively.

**Table 4.** The 20 most cited journals in land consolidation and rural revitalization.

| Sources | Number of Citations |
|---|---|
| LAND USE POLICY | 2932 |
| J RURAL STUD | 779 |
| HABITAT INT | 626 |
| J GEOGR SCI | 502 |
| LANDSCAPE URBAN PLAN | 340 |
| SUSTAINABILITY-BASEL | 275 |
| WORLD DEV | 259 |
| JOURNAL OF GEOGRAPHICAL SCIENCES | 257 |
| SCI TOTAL ENVIRON | 229 |
| APPL GEOGR | 214 |

**Table 4.** *Cont.*

| Sources | Number of Citations |
| --- | --- |
| TRANSACTIONS OF THE CHINESE SOCIETY OF AGRICULTURAL ENGINEERING | 204 |
| J ENVIRON MANAGE | 200 |
| AGR SYST | 194 |
| NATURE | 188 |
| ECOL INDIC | 178 |
| P NATL ACAD SCI USA | 155 |
| SCIENCE | 153 |
| AGR ECOSYST ENVIRON | 145 |
| J CLEAN PROD | 144 |
| J PEASANT STUD | 140 |

### 4.4.2. Analysis of Main Authors' Historical Citations

Based on the R language, we visually analyzed the historical quotations in the field of land consolidation and rural revitalization by using historical quotations in the Bibliometrix installation package. Overall, 14 nodes were selected, and the content of each node was set as the author and the year of publication (Figure 6). In the visual analysis of historical citation, LCS (local cited document) and GCS (global cited document) are two important indicators. Whilst the LCS refers to the reference in the downloaded thesis data set, and GSC refers to the reference in the scientific core database.

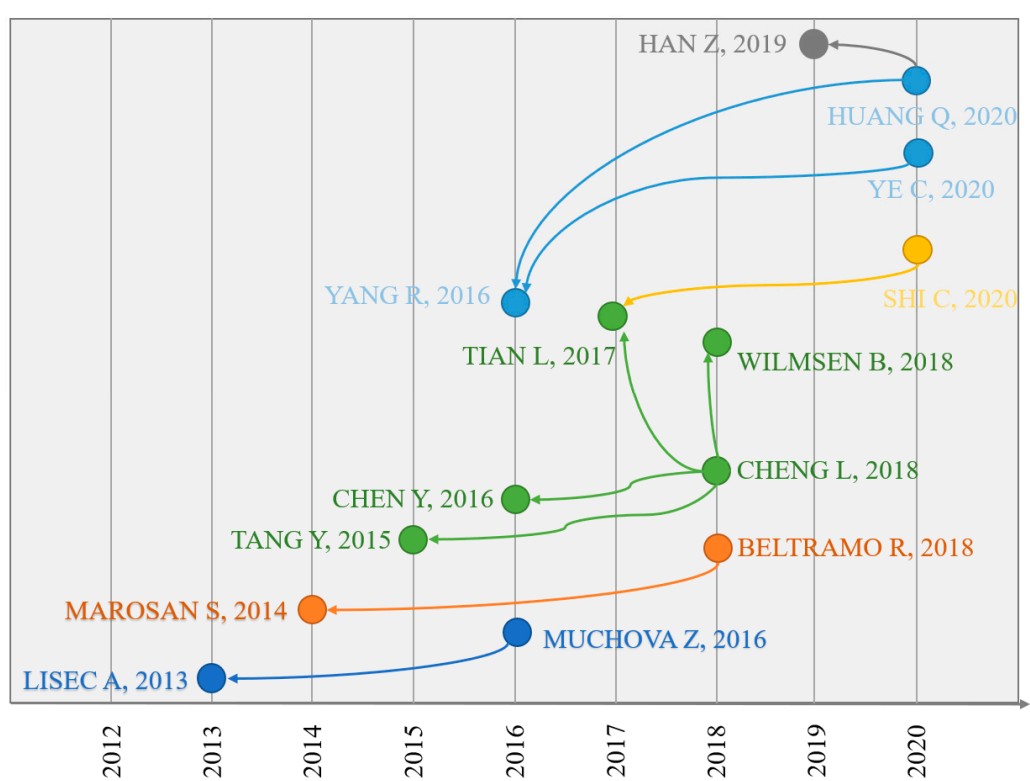

**Figure 6.** The historical direct citation network of the top 10 cited papers in the field of land consolidation and rural revitalization from 1950 to 2021.

As seen in Figure 4, the earliest node from 2012 to 2021 is an article published in GEODETSKI VESTNIK in 2013, entitle "The state of the art and challenges in the field of agricultural land consolidation in Slovenia". Several classical articles were published from 2012 to 2020, clearly showing the historical citation relationship between the top 10 cited authors in this field. For example, Yang published "Spatial distribution characteristics and optimized reconstruction analysis of China's rural settlements" in the JOURNAL OF

RURAL STUDIES in 2016. The article "The process of rapid urbanization" has two different citation chains (Figure 6), and its LCS is 26 (Table 5), with a GCS of 113 (Table 6). It is also the article with the highest LCS and GCS ranking from 1950 to 2021. In this study, the spatial distribution characteristics of county-level rural areas in China were simulated by geographic detector research methods, the differences in the spatial distribution among different regions were analyzed, and the spatial distribution characteristics and patterns of rural areas in China were summarized to provide a research foundation for the optimization of rural spatial layout and the rational allocation of resources in China.

**Table 5.** The top 10 literatures of LCS in the field of land consolidation and rural revitalization.

| Documents | DOI | Year | LCS | GCS |
|---|---|---|---|---|
| YANG R, 2016, J RURAL STUD | 10.1016/j.jrurstud.2016.05.013 | 2016 | 26 | 113 |
| TANG Y, 2015, J RURAL STUD | 10.1016/j.jrurstud.2015.09.010 | 2015 | 15 | 37 |
| MUCHOVA Z, 2016, ECOL ENG | 10.1016/j.ecoleng.2016.01.018 | 2016 | 11 | 37 |
| TIAN L, 2017, URBAN STUD | 10.1177/0042098015615098 | 2017 | 7 | 31 |
| OLDENBURG P, 1990, WORLD DEV | 10.1016/0305-750X(90)90047-2 | 1990 | 6 | 18 |
| CHENG L, 2018, HABITAT INT | 10.1016/j.habitatint.2018.04.004 | 2018 | 4 | 28 |
| HAN Z, 2019, J CLEAN PROD | 10.1016/j.jclepro.2019.117888 | 2019 | 3 | 31 |
| HUANG Q, 2020, J CLEAN PROD | 10.1016/j.jclepro.2020.123150 | 2020 | 3 | 8 |
| HESTON A, 1983, EXPLOR ECON HIST | 10.1016/0014-4983(83)90022-0 | 1983 | 2 | 28 |
| MAROSAN S, 2014, GEOD VESTN | 10.15292/geodetski-vestnik.2014.03.568-577 | 2014 | 2 | 2 |

**Table 6.** The Top 10 Literature of GCS in Land Consolidation and Rural Revitalization.

| Paper | DOI | Year | LCS | GCS |
|---|---|---|---|---|
| YANG R, 2016, J RURAL STUD | 10.1016/j.jrurstud.2016.05.013 | 2016 | 26 | 113 |
| TANG Y, 2015, J RURAL STUD | 10.1016/j.jrurstud.2015.09.010 | 2015 | 15 | 37 |
| MUCHOVA Z, 2016, ECOL ENG | 10.1016/j.ecoleng.2016.01.018 | 2016 | 11 | 37 |
| TIAN L, 2017, URBAN STUD | 10.1177/0042098015615098 | 2017 | 7 | 31 |
| HAN Z, 2019, J CLEAN PROD | 10.1016/j.jclepro.2019.117888 | 2019 | 3 | 31 |
| CHENG L, 2018, HABITAT INT | 10.1016/j.habitatint.2018.04.004 | 2018 | 4 | 28 |
| HESTON A, 1983, EXPLOR ECON HIST | 10.1016/0014-4983(83)90022-0 | 1983 | 2 | 28 |
| SHI C, 2020, URBAN STUD | 10.1177/0042098019845527 | 2020 | 1 | 20 |
| CHEN Y, 2016, HABITAT INT | 10.1016/j.habitatint.2016.01.002 | 2016 | 2 | 19 |
| WILMSEN B, 2018, DEV CHANGE | 10.1111/dech.12372 | 2018 | 2 | 19 |

The top three articles in LCS and GSC in the field of rural revitalization (Tables 5 and 6) focused on land consolidation. The article "China's rural settlements" was published by Yang in the JOURNAL OF RURAL STUDIES in 2016. During a period of rapid urbanization, "Tang published the article" "Governments' functions in the process of integrated consolidation and allocation of rural-urban construction Land in China in the JOURNAL OF RURAL STUDIES, and the publication "Possibilities of optimal land use as a consensus of lessons learned from land consolidation projects" appeared in the JOURNAL OF ECOLOGICAL ENGINEERING" in 2016. These three articles are high-level publications in the field of land consolidation and rural revitalization, with LSC values of 26, 15, and 11 and GCS values of 113, 37, and 37, respectively. Among them, the second article discusses the key issues of the comprehensive integration and allocation of urban and rural construction land in China and puts forward the viewpoint that "the government functions need to be better integrated to solve problems such as the great difference in the social and economic development levels in metropolitan areas and the different wishes and needs of relevant stakeholders", providing a solid foundation for the future research of urban and rural land consolidation.

## 4.5. Keyword Analysis

Keywords summarize the topics and contents of research at a high level. High-frequency keyword analysis directly indicates the hot spots in any research field. The word tree of the top 20 keywords in the field of land consolidation (Figure 7) was drawn using Bibliometrix. Based on the results, land consolidation, policies, and fragmentation were the most frequent keywords, indicating that land fragmentation is the main issue in the field of land consolidation, and policy measures play a key role in solving this issue. Among the keywords, "China" appeared 50 times, ranking 5th, indicating that China is one of the main regions in terms of research in land consolidation and rural revitalization. This is in line with the total publication number and the international cooperation of China analyzed above. The keywords "influence" and "driving factors" appeared 38 and 36 times, respectively.

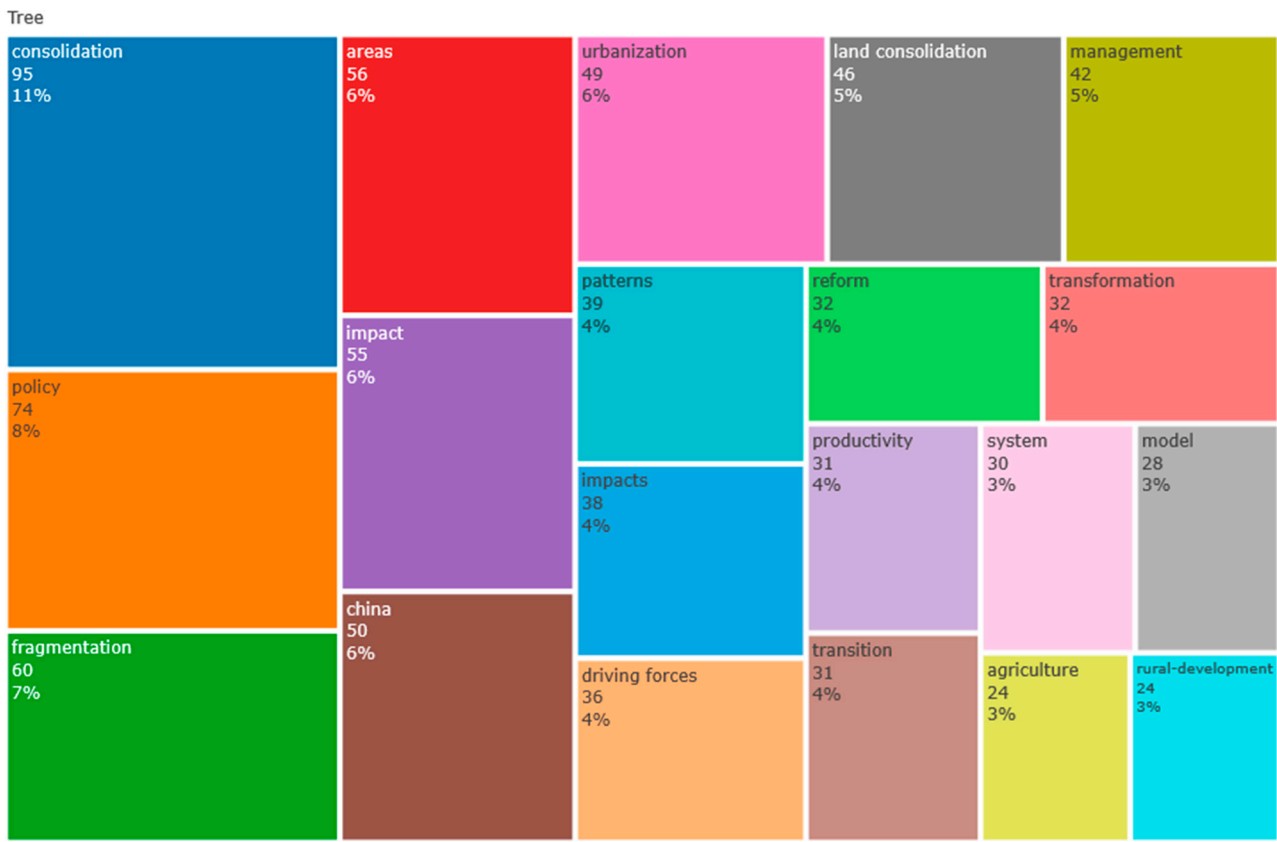

**Figure 7.** Word tree of keywords in land consolidation and rural revitalization studies from 1950 to 2021.

Clustering analysis is an important visual analysis in bibliometrics. Its principle is based on the frequency of two keywords appearing at the same time, and the complex keyword network relationship is simplified into several small groups by statistical methods. The main purpose of clustering analysis is to classify clusters based on similar keywords and to minimize the similarity between clusters. In this study, keyword cluster analysis in the Bibliometrix software package was used to classify and analyze multiple keywords, with the aim to cluster keywords with high similarity to the same cluster. Finally, all individuals were grouped into one category, and the similarity of keywords in the research fields of land consolidation and rural revitalization is shown in the form of a tree diagram (Figure 8).

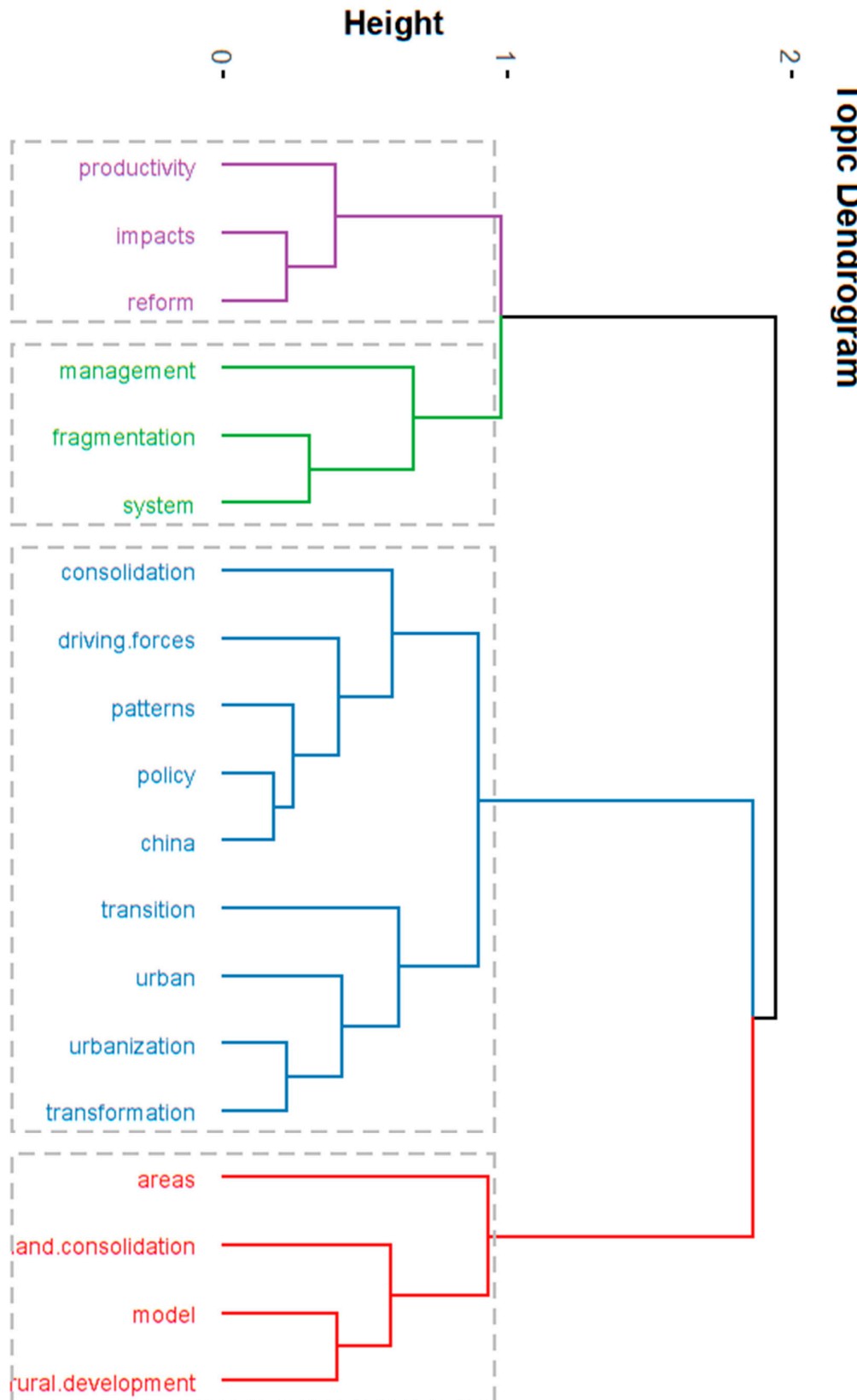

**Figure 8.** Tree diagram of systematic cluster analysis of keywords in land consolidation and rural revitalization from 1950 to 2021.

As seen in Figure 9, we obtained four different clusters.

(1) Technical efficiency and agricultural production, reflecting issues in the implementation of land consolidation projects.
(2) Impacts of policy management, systems, and land fragmentation.
(3) Impacts of land consolidation on land use transition and its driving factors against the background of accelerated urbanization.
(4) Performance evaluation of land consolidation, adoption of models, and rural development.

**Figure 9.** Multiple correspondence analysis of high−frequency keywords in land consolidation and rural revitalization from 1950 to 2021.

### 4.6. Topic Analysis

Studying the theme and theme evolution of a field is conducive to understanding the research progress in this field. The Sankey diagram, also known as the Sankey energy diversion diagram or the Sankey energy balance diagram, is a specific type of flow chart in which the width of the extended branch corresponds to the size of data flow; it is generally applied in the visual analysis of data such as energy, material composition, and finance data. The most obvious feature of the Sankey diagram is that the sum of the branch widths at the beginning and end is equal, that is, the sum of all the main branch widths should be equal to the sum of all branch widths to maintain energy balance. In the network, the Sankey diagram can visualize the traffic between nodes. Because of its wide practicability, it has been applied in many geographical or human environmental studies. Based on the visualization of the time-varying themes in the field of land consolidation and rural revitalization, our study shows the diversion of different themes and provides quantitative information, such as on theme flow and conversion.

According to Figure 8, the topics of land consolidation and rural revitalization largely changed during the period from 1950 to 2021, and we analyzed the topic changes with time nodes of 2006 and 2015. From 1950 to 2021, the theme in the field of land consolidation showed four evolution paths (Figure 10).

(1) Research on land fragmentation. The first path is conservation → impact → fragmentation, consolidation, and the second path is land, farmers, climate change → fragmentation. Land fragmentation is an important research topic in land consolidation and runs through the whole process of land consolidation development. In rural land consolidation, the fragmentation of cultivated land is the main issue that needs to be solved. Cultivated land fragmentation not only enriches agricultural planting structure, disperses agricultural planting risks, and increases farmers' income but also causes the waste of land resources and the increase in agricultural production costs, thus reducing agricultural production efficiency and impeding agricultural mechanization. Land consolidation is an effective way to reduce the fragmentation of cultivated land. Ela [72] examined the effects of land consolidation projects on the fragmentation of arable land, using Turkey as an example. Prior to land consolidation, there were 1.17% and 3.7% of agricultural enterprises, with an index value below 0.40; after land consolidation, there were 0.6% and 2.3% of agricultural enterprises. The findings indicate a decrease in land block fragmentation in this region. Land consolidation also greatly benefited the local agricultural business owners financially.

(2) Research on the development process of land consolidation. The first path is management → growth → fragmentation, consolidation, the second path is region → policy → fragmentation, consolidation, and the third path is system → fragmentation, consolidation. Land consolidation developed as early as the 14th century, with a history of several hundred years. However, the development process and goals of land consolidation differ among countries. Understanding the development process of land consolidation is conducive to analyzing the land problems in the development process of various countries and adopting relevant policies. In Germany, land consolidation is concentrated on building rural infrastructure, transforming regional planning, and protecting natural landscapes. Public involvement in land consolidation projects is also given special consideration. Previously, the focus of land consolidation in Germany was on adjusting farmland ecology and enhancing agricultural supporting facilities [73]. In the Netherlands [74], land consolidation is currently focused on enhancing the land's overall function and bolstering the preservation and enhancement of the natural environment. In Japan, land consolidation initially sought to regulate the salinization of agricultural land and land reclamation and later evolved into overall planning. Land consolidation has been a crucial tool for achieving the synergy between residential space and agricultural land since the 1980s [75]. The preparation of detailed development plans for rural communities with an agricultural focus clearly demonstrates the significant role of land consolidation in rural development.

(3) Impact of land consolidation on living things. The first path is bioremediation → impact → fragmentation, consolidation, and the second path is biodiversity → fragmentation. First, in land consolidation, large-scale projects need to be carried out. With such projects, the living environment of animals and plants is destroyed, the habitat of organisms is reduced, biodiversity is declined, and the ecological balance in a given region is disturbed. Second, the soil is an important object of land consolidation, and the soil microbial community is greatly affected by land consolidation. The theme evolution direction of land consolidation's impact on biology is mainly the impact of land consolidation on soil physical structure, nutrient cycling, and microbial functions. This research is of great significance for avoiding the negative effects of land consolidation and improving its benefits. For example, He [76] described the soil microbial communities under five land use patterns and used DNA fingerprint and metabolic analysis, revealing that land reclamation seriously affected the soil microbial communities qualitatively and quantitatively.

(4) Research on the methods and models of land consolidation. The path is model → impact → fragmentation, consolidation. The model method of land consolidation is an important branch of land consolidation research and the foundation of land

consolidation development. With the gradual maturity of the model method, the content of land consolidation research is enriched. The period before 2005 was the most important period regarding the development of models and methods. After 2006, model methods were gradually applied to practical research. For example, Zhang [77] et al. introduced resilience thinking into land remediation, constructed the conceptual model of "land remediation+ecology", designed the path of resilience construction in land remediation, and opened up a new direction and foundation for the theoretical innovation of land remediation in the new era.

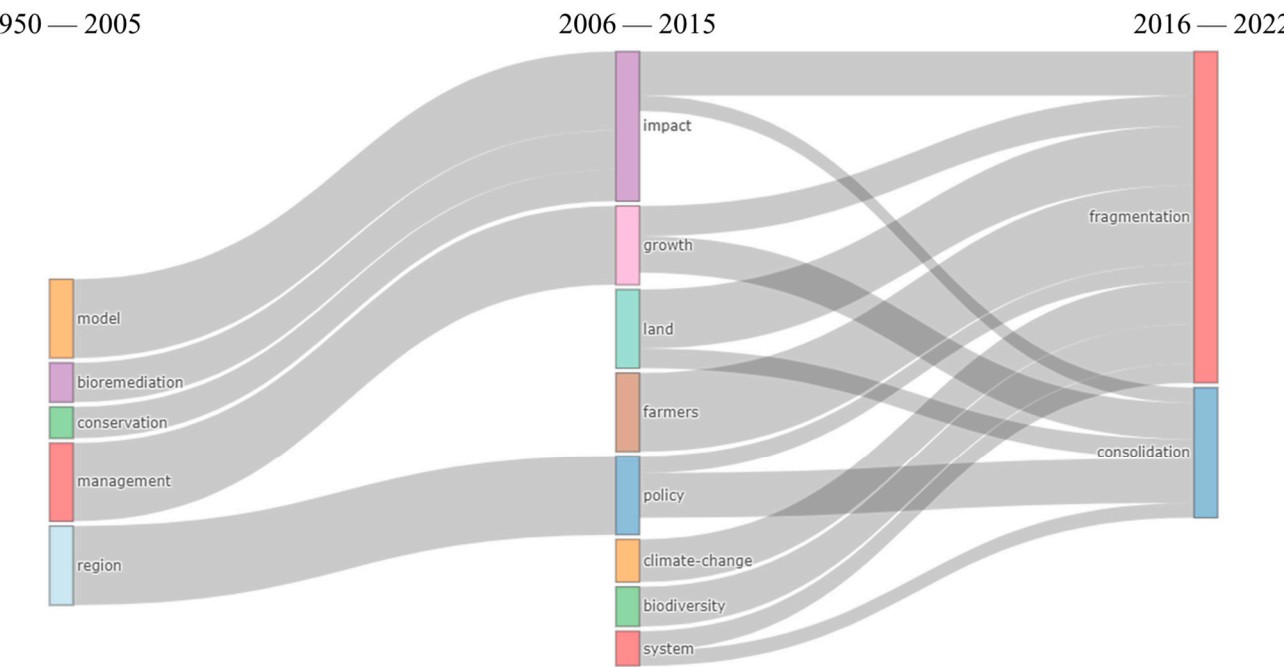

**Figure 10.** Evolution of topics in land consolidation and rural revitalization from 1950 to 2021.

## 5. Discussion

### 5.1. Development of Land Consolidation Theory

Land consolidation can be dated back to the 14th century, and related legislation was established in the middle of the 18th century [78]. Traditionally, it is the most favorable cultivated land management method to solve the issue of land fragmentation and improve land treatment efficiency. Over the years, it has been transformed into a modern practice, and its multiple objectives cover sustainable agriculture, rural development, ecological environment protection, and sustainable land management [39]. Land consolidation has been applied in many countries in the world, and its content varies largely among countries.

#### 5.1.1. Practical Analysis of Germany and The Netherlands

From the Middle Ages, when productivity was relatively low, through the beginning of the 20th century, the land was controlled by "little pieces and huge pieces", and scattered small pieces of land were combined to improve the collective benefits of the land. Land consolidation in the 1930s, while the economy was booming, concentrated on reserve land, readjusting and planning the land that had become haphazardly dispersed due to infrastructure development. After this, the European economy was growing, society was thriving, public security was steady, and people's yearning to reconnect with nature was at an all-time high in the 1970s [79]. During this period, the goal of land consolidation was to safeguard the ecological balance of the terrain. The effectiveness of land consolidation in Germany is a result of land consolidation-related legislation and regulations, ideal institutions for land consolidation, ownership management during the land consolidation process, and significant public involvement [80].

The Netherlands is among the nations in which land consolidation has been playing an important role for a long time. To facilitate agricultural automation production, scale planting, and advanced agricultural growth, the country started to engage in low-level land consolidation at the beginning of the 20th century, mostly by merging scattered land and changing the planning [81]. The first Land Consolidation Law was enacted in the Netherlands in 1924, with the goals of centralizing land management, increasing agricultural mechanization, and raising agricultural output levels. The second Land Consolidation Law, which served the same function as the first one, was enacted in 1938 to simplify the land consolidation processes. The country's land consolidation has been advancing since the 1940s [82]. Particularly, the 1954 Land Consolidation Law represents a significant turning point, with the advent of rural landscape planning. By planning the arrangement of the rural landscape, landscape architects, as opposed to civil engineers in the past, focused on rural aesthetics and transformed the ecological aesthetics of the Dutch rural landscape.

5.1.2. Analysis of China's Practice

China has experienced four periods of land consolidation. In the first phase from 1986 to 1997, there was no land consolidation for construction, and land consolidation was mostly applied in terms of farming and reclamation [83]. Land consolidation and reclamation, which began in 1998 and finished in 2007, was the theme of the second phase. With China's rapid industrialization and urbanization since the mid-1990s, more land is being used for construction and less for agriculture. A total of 3.9106 hm$^2$ of land was added to the construction sector between 1996 and 2008, whereas 1.5107 hm$^2$ of cultivated land was shifted to other land use types. During this time, land consolidation was focused on the expansion of arable land, the decrease in land fragmentation, and the increase in agricultural productivity [33]. The multi-type land consolidation stage from 2008 to 2012 represents the third stage of land consolidation. During this period, the scope of land consolidation expanded to include damaged land in urban and rural areas as well as idle and abandoned industrial and mining sites. Increasing cultivated land, reducing fragmentation, enhancing agricultural infrastructure and environmental quality, and raising agricultural productivity are the overarching objectives of land consolidation [84]. The complete land consolidation stage, the fourth step, started in 2013. Comprehensive land consolidation is defined as an activity that systematically corrects the land, waters, roads, forests, and villages in an area to increase agricultural production levels and promote agriculture, rural population concentration, industrial agglomeration, and urban-rural integration [40]. Land consolidation development demonstrates how important it is for boosting the quantity and quality of arable land, enhancing the natural environment, and fostering the coordinated growth of urban and rural areas. However, in China, in the new period, particularly against the backdrop of the development of an ecological civilization, the components of the long-standing policy of land consolidation need to be carefully addressed.

*5.2. Analysis of the Changes in the Number of Publications and Citations in Land Consolidation and Rural Revitalization*

The number of publications on land consolidation and rural revitalization is currently increasing. Although land consolidation has a long history, before the 20th century, only a few studies on this issue were published, and land consolidation was not adopted as an authoritative tool by many countries. It was not until the 1980s and 1990s, that a mature model of land consolidation gradually formed and became a popular agricultural policy in Asia and Eastern Europe [26,85]. From the end of the 20th century to the beginning of the 21st century, research on land consolidation developed slowly. Since the beginning of the 21st century, the FAO has been publishing a series of manuals to promote international cooperation and guide land consolidation projects around the world [39]. Land consolidation research has been developing rapidly, and the number of publications has

increased. Consequently, land consolidation methods have been adopted gradually in land management.

Before 1996, the citation rate of articles was generally low, and research in this field was in its infancy. There were few respective studies, and the number of citations was low, sometimes reaching zero. During 1997–2004, the citation number was high, and land consolidation attracted widespread interest. Most articles published during this period had a "pioneering character" [10] and provided a basis for subsequent research. Therefore, they were highly cited. After 2005, the average citation rate of papers spiraled but was lower compared to that of the previous period. During this period, land consolidation became a hot topic all over the world, and the number of publications increased gradually. The research interests were complex, which led to a large fluctuation in the number of citations.

### 5.3. Land Consolidation and Rural Revitalization

Responding to the loss and decline in rural internal factors, stimulating internal power, and absorbing outside resources through economic, political, and cultural construction, as well as recombining rural population, land, and industry are all examples of rural revitalization [21,86]. The goal is to optimize the structure of factors, enhance regional functions, reshape rural morphology, and realize the overall economic, social, and ecological revival of rural areas as well as a new pattern of urban-rural integration. The interaction of numerous different elements creates the open system known as the rural system, which has a comprehensive multidimensional and dynamic evolution [87]. The three main factors that influence rural social and economic development are population, land use, and industry.

#### 5.3.1. Connotation Expansion of Land Renovation from the Perspective of Rural Revitalization

In rural areas, land is both the most abundant resource and the primary factor in production. Activating land as a valuable factor resource, optimizing the spatial layout for rural agricultural production, village construction, and industrial development, accurately allocating land for all types of industries, speeding up construction, and promoting the implementation of "multi-regulation integration" in rural areas based on land policy and village land use planning are important factors.

The traditional definition of rural land remediation is more restricted to the technical engineering components, and its primary goals center on the material features, such as increasing the size of farmland, enhancing the quality of cultivated land, and improving the village layout [88]. Land cleanup is generally regarded as a way to make room for urban development. In the context of the new normal, rural revitalization is seen as improving living conditions and public services as well as deeply reviving thriving industries, distinctive cultures, and a well-functioning political system. The multi-functional value of rural areas and the differences between urban and rural regional systems should serve as our foundations for achieving comprehensive rural rejuvenation [7]. We should avoid replicating the linear transformation processes of "rural industrialization" and "rural urbanization" and instead pursue sustainable connotative development. From the viewpoint of rural revitalization, land remediation has new meanings and purposes:

(1) To obtain the essential components of rural development–population, land, and industry–moving. Our goal is to realize the coordinated coupling of rural "population-land-industry" by promoting the reconstruction of production, living, and ecological space, strengthening organic integration with diverse rural formats such as modern agriculture, experience agriculture, homestay management, and tourism, promoting the non-agricultural transfer of rural population, and changing the land use mode [15,21].

(2) To coordinate the revival of the physical world. Based on comprehensive land improvement, we should alter the quantity and quality of cultivated land as well as the pattern of use of rural construction land [21,86,89]. We also need to revitalize rural land resources, protect traditional village features, pass down local cultural values,

continue the settlement style, maintain the distinct charm of villages, and improve the ecological and cultural functions of rural areas.

(3) Reorganization of the rural governance system and the docking area. With the aggregation of rural living space, we should investigate the rural governance mode of combining local government, grassroots organizations, and communities, strengthen the development of a new class of business subject-capable individuals who are suitable for moderate-scale operations, and realize the effective connection between rural space and organizational governance system reconstruction [12,90].

### 5.3.2. Internal Drivers of Rural Revitalization

As a comprehensive and methodical national plan, rural regeneration entails the peaceful coexistence of a number of interconnected and mutually exclusive elements, including economic, political, cultural, social, and ecological ones. Land remediation is a methodical process that involves the comprehensive repair of "fields, water, roads, woods, and villages". It is driven by a variety of internal elements, including economic, social, and political issues, and is a key platform for rural revitalization [91,92].

(1) The economic drivers of rural revitalization to land remediation

First, there is a significant increase in demand for land. With accelerated industrialization and urbanization, the need for land is rising from the standpoint of industrial development. We cannot resolve the conflict between "guaranteeing development" and "guaranteeing the red line" without intensifying land consolidation. Both industrialization and urbanization demand huge land areas [93]. Second, the land use is vast, and the reserve resources are insufficient. Farmland yield has been significantly impacted by the phenomenon of low-intensity management and abandonment, as well as by land pollution and soil erosion. Only through boosting land cleanup will we be able to increase grain production, enhance farmland quality, and encourage the rapid growth of the agricultural economy. Third, there are limitations to agricultural land use [94]. Land remediation can increase the size of the land and the operation of the cultivated land, change the original production mode, and, at the same time, combined with the mode of land circulation, improve the efficiency of production, free up labor, and give farmers access to other avenues to increase their income.

(2) The social drivers of rural revitalization to land remediation

First, there are many social paradoxes in rural areas. Inconsistencies in village finances and cadre behavior, issues with "old weakening" and "empty villages," and flaws in the rural social security system are only a few examples [95,96]. These issues and inconsistencies are detrimental to social stability because they impede the development of community culture and increase farmers' social living standards. Second, there is uneven development in urban and rural areas. The significant income disparity between urban and rural populations is the key indicator. In addition to having lower incomes than urban inhabitants, rural dwellers also experience a weak social welfare security system, limited access to subsidies, and a dearth of fundamental public services [97]. As a result, it is essential to boost agricultural output, raise farmers' incomes, gradually raise the quality of basic public services in rural areas, close the gap between urban and rural areas, and encourage urban-rural integration.

(3) The political drivers of rural revitalization to land remediation

The party and the government prioritize and are accountable for rural rehabilitation. They actively pursue the government's rural rehabilitation agenda and assume a leadership and directing position in land remediation initiatives. Leading departments must direct land consolidation initiatives toward rural regeneration through financial and legal means, spend funds wisely, combine forces, fulfill their overall responsibilities, and enhance the effectiveness of land consolidation [98,99]. To facilitate rural revitalization strategies, land remediation must be put into practice. The political power of the rural working class is

also significantly diminished. On the one hand, the quality of rural administrative subjects has deteriorated significantly, the rural elite population is seriously declining, and the aging issue of party members results in a shortage of resourceful and competent grassroots cadres in rural areas [100]. On the other hand, the poor financial foundation in rural areas limits the capacity of local governments to implement policies, which lowers farmers' confidence in the government. Third, many rural emergencies cannot be handled in a timely manner due to a lack of funding and the waning popularity of grass-roots party groups, severely weakening the capacity for social mobilization [87,101]. Government departments with strong execution capabilities, ample funding, and credibility must prepare and make decisions about the implementation of land repair. Implementing land cleanup can boost farmers' income while also uniting them, consolidating rural political power, and fostering ties between the population and the government.

### 5.3.3. Mutual Feedback Relationship between Land Remediation and Rural Revitalization

Rural regional systems will experience varying degrees of feedback and response during rural development and transformation, depending on how quickly or slowly urban and rural populations are changing, as well as other socioeconomic development factors, including abnormal evolution [21,89]. This will have an impact on the long-term sustainability of regional agriculture and rural areas. The economic benefits of agricultural production and the industrial chain linked by agricultural products are directly impacted by landscape optimization, quality improvement, or the facility improvement of land resources because they serve as the spatial carrier of the main social and economic activities. This, in turn, affects the income level and employment preferences of rural residents, changing the industrial structure and human resource structure of the rural area [102]. Increasing the area that can be productively farmed, improving the conditions for agricultural production and the quality of farmed land, promoting the industrialization of agriculture and its scale, or creating rural tourism based on the enhancement of farmland landscape are the main objectives of agricultural land remediation [37,85,99]. Through scientific planning, it is possible to direct population and industrial concentration on rural construction land, realizing the benign interaction between urban capital and rural idyll and vacant land resources, and opening up new ways and channels for capital and resources flow between urban and rural areas [32].

The fundamental goals of land remediation are to change property ownership and organize land use, whereas the adaptability of land resources provides ancillary benefits, such as coordinating urban and rural growth, preserving the character of the landscape, and maintaining social stability [20,103,104]. To achieve the transition from "land-based" single-element regulation to "people, land, and industry" multi-element coordinated coupling comprehensive remediation, land remediation should organically integrate new formats, new technologies, new subjects, and other elements [105]. This is because rural areas face extensive resource use, lagging facilities, accelerated element loss, and weak management. The focus on land cleanup will eventually shift from quantitative growth to rural areas [106].

### 5.4. Limitations

We analyzed the research status, characteristics, regional differences, and evolution laws of land consolidation and rural revitalization based on publications in the Web of Science database. However, we could not determine the relationship between land consolidation and rural revitalization due to the selection of search terms when accessing data and the restriction of research methods. Studies on the effects of land consolidation on rural revitalization have achieved certain results. For example, Fang et al. [107] analyzed the means and purposes of land consolidation practices as well as villagers' attitudes and behavior through a comparative case study of two villages. Rao [108] analyzed cases in which land consolidation promoted rural transformation in towns from the perspective of socio-economic transformation. However, it is still challenging to carry out quantitative

analysis in this field. In the future, the methods and systems should be optimized to quantify the impact of land consolidation on rural revitalization.

## 6. Conclusions

The period from 1950 to 2021 saw an evolution of research on land consolidation and rural revitalization. During this period, the number of publications per year, the number of citations per year, and topics changed greatly. In the early stage, most studies focused on keyword co-occurrence analysis, journal sources, and published papers, whereas few studies considered the context of historical citations, high-frequency keyword cluster analysis, discipline evolution, and the prediction of future development directions. Therefore, based on bibliometrics and the Bibliometrix software package, we conducted a visual analysis of the research status, characteristics, regional differences, and evolution laws of land consolidation and rural revitalization. From 1950 to 2021, the number of research publications on land consolidation and rural revitalization fluctuated slightly, albeit with a general increase. A total of 1715 authors published articles on land consolidation and rural revitalization from 1950 to 2021. China and the United States were the leading countries in terms of international cooperation.

"Land consolidation", "policy", "fragmentation" "region", and "China" were high-frequency keywords in this field in recent years. At the same time, the research on the development of land consolidation is important to the research on land consolidation and rural revitalization. China is the largest and most populous developing country in the world and has a long history of rural development. Against the background of rapid urbanization, China's rural revitalization presents numerous problems, and land consolidation is an important tool to solve them and promote rural revitalization. From 1950 to 2021, the field of land consolidation and rural revitalization experienced three evolutionary stages. Land fragmentation was a constant issue, and the main themes in the field of land consolidation and rural revitalization have evolved into "land fragmentation" and "protection", which indicates that the core of land consolidation has changed into solving the issue of land fragmentation and ensuring comprehensive and sustainable rural development. Rural revitalization is in line with new requirements for land consolidation, and land consolidation has become a comprehensive means to comprehensively consider social-economic-political factors. However, whilst land consolidation optimizes rural spatial layout and improves rural land use efficiency, it also brings some problems. For example, land consolidation will further aggravate the occupation of rural land, resulting in more cultivated land being occupied, and may therefore infringe the interests of some landowners. Future research in the field of land consolidation and rural revitalization should focus on the feedback mechanism of land consolidation and rural revitalization, land consolidation, and farmland encroachment, how the government can protect the rights and interests of all landowners, and how to realize the coupling of multiple factors, thus making land consolidation an effective management tool for coupling social, economic, ecological, and policy factors and facilitating rural revitalization.

**Author Contributions:** S.L.: Methodology, Software, investigation, validation, Data curation, Writing—original draft; W.S.: Conceptualization, funding acquisition; S.L. and W.S.: Writing, proofreading—review & editing. All authors have read and agreed to the published version of the manuscript.

**Funding:** This research was funded by The Second Tibetan Plateau Scientific Expedition and Research grant (Grant No. 2019QZKK0603) and the Project of National Natural Science Foundation of China (grant number 42071233).

**Data Availability Statement:** Here no new data were created.

**Conflicts of Interest:** The authors declare no conflict of interest.

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
