# Peer review of "Research Progress in Land Consolidation and Rural Revitalization: Current Status, Characteristics, Regional Differences, and Evolution Laws"

_land, doi:10.3390/land12010210_

Round 1
Reviewer 1 Report
Overall this paper has some interesting information on an important topic and overall I would like to see this published with some major changes. At the moment it provides a basic review of the quantity of work undertaken on rural land consolidation and also a very limited thematic analysis. If the paper were to expand the thematic analysis is would be a more robust and useful paper. My suggestion is that this paper be reviewed to revise and extend the discussion to document the major theoretical contributions over time. This analysis should highlight the seminal publications that helped shape the overall body of work. It should also look at how understandings developed in one context - for example China or Germany - have been used by academics to understand and theorise rural land consolidation.
The discussion should then also be revised to again draw out the main theories that have been developed.
Finally the conclusion should highlight the most contemporary dimensions of the research undertaken in the last few years and highlight what gaps in knowledge remain.
If these major revisions are undertaken, this paper could become a highly useful and citable paper that will help move forward research on rural land consolidation.
Author Response
Thank you for your comments. We have responded to your comments point by point. Please refer to the attachment.

Reviewer 2 Report
This paper reviews the research progress of China’s land consolidation and rural revitalization. The visualisation of your review results is great, and I enjoy watching your graphs.
You made a good start but still need lots of effort to make it publishable.
First, the review has a bit shallow and biased understanding of land consolidation. You didn’t even realise land consolidation could be ending up with land grabbing. You only see land consolidation as tool/policy instrument, which is positive in general. This viewpoint is a bit away from the scholarship’ views on land consolidation. Successful consolidation must overcome competing interests of the land holders or users. Your article downplays this and therefore most of your analysis is a bit floating on the surface.
Second, conducting a literature review is not hard but writing a good review paper is not easy. Sometimes it will be extremely hard if you choose to review literatures on a contentious topic. Unfortunately, land consolidation is exactly such a topic. You need to think of what new knowledge could be generated by reviewing old knowledge. I don’t think such a bibliometric paper like you wrote on current format generates new knowledge. What’s worse, it may confuse starters in this field and view land consolidation as a silver bullet, as described by your paper. To review such a complicated topic, you need a lot of preparation.
To improve, considering:
1) Find a research question to guide your review
2) Please tell us clearly what you want to achieve
3) Try to view land consolidation as a progress rather than a tool; you could think about: who are involved? Who are affected? Who are winners and who are losers? what are the purposes of land consolidation?
4) Follow a well-developed review approach, could be either scoping or systematic review. Follow each step rigorously.
Author Response

(The authors gave the same response as above.)

Reviewer 3 Report
Overview
Overall, the article reviews the research results of Land Consolidation and Rural Revitalization using a recent and popular bibliometric approach. However, the article is confusing to read. is Land Consolidation and Rural Revitalization one or two keywords? If it is one keyword, then the relationship between the two should be the focus of bibliometric attention. If it is two keywords, then the article content does not match. Authors need to reorganize the logic of their articles, and the following suggestions may be helpful to authors for future publication.
Problems and Recommendations
General
1. Are there two parallel keywords in the title? The ambiguity of the title makes it difficult for readers to understand the content of your article.
2. Lines 62-67, regarding the definition of land consolidation, need to be redefined. Land rights are the basis for the realization of land functions.
3. Include article structure in the introduction.
4. In the Data source section, the keywords retrieved were literature containing both keywords, but the subsequent analysis did not discuss scholarly research on the relationship between the two, which is the biggest problem with the article in my opinion.
5. Adjust the title font in Figure 2, Figure 3, and Figure 4.
6. The authors highlight the outstanding contributions of the article as the analysis of citations, keyword clusters and topic evolution, but the analysis is not detailed enough, especially for the topics. Also, it is recommended that these three core points be arranged consecutively.
7. Why is the discussion section missing a discussion of keyword clustering and topic evolution? What is the relationship of section 5.3 to the content of the article?
8. In the Conclusion section, no implications are given for this article.
Miscellaneous
It is recommended to re-read the text to eliminate small typos.
Author Response

(The authors gave the same response as above.)

Round 2
Reviewer 3 Report
The paper is accepted in its form after the revision.